# Clinical acuity and National Early Warning Scores (NEWS2) of remotely monitored patients on virtual wards: A retrospective cohort study

Lauren J. Scott[1,2]*, Rebecca Winterborn[3,4], Joanne Appleton[5], Chris Penfold[1,2], Jenny Tomkinson[4], Isobel Ward[1,2], Frank de Vocht[1,2], Alison Tavaré[5,6]

1 National Institute for Health and Care Research Applied Research Collaboration West (NIHR ARC West), University Hospitals Bristol and Weston NHS Foundation Trust, Bristol, United Kingdom, 2 Population Health Sciences, Bristol Medical School, University of Bristol, Bristol, United Kingdom, 3 North Bristol NHS Trust, Bristol, United Kingdom, 4 Sirona Care & Health, Bristol, United Kingdom, 5 NHS England South West, Bristol, United Kingdom, 6 Health Innovation West of England, Bristol, United Kingdom

* Lauren.Scott@bristol.ac.uk

## Abstract

### Background

Virtual wards are an NHS priority, designed to deliver acute care, monitoring and treatment to people at home, providing an alternative to hospital admissions or facilitating earlier hospital discharge. The aim of this study was to understand the clinical acuity, care pathways and outcomes of people admitted to virtual wards in Bristol, North Somerset and South Gloucestershire (BNSSG) who were remotely monitored.

### Methods

A retrospective observational cohort study of all remotely monitored patients aged 16 + years admitted to a virtual ward in BNSSG between October 2023-February 2025. Clinical observations (respiratory rate, oxygen saturations, systolic blood pressure, pulse rate and temperature) were collected, and National Early Warning Scores (NEWS2) were calculated. The area under the curve (AUC) of NEWS2 to predict hospital (re)admissions is presented.

### Results

2,533 admissions across five care pathways were included: respiratory (41%), frailty (18%), outpatient parenteral antimicrobial therapy (19%), heart failure (7%) and general (18%; pathways not mutually exclusive). Median virtual ward length of stay was 10 days (interquartile range 6−14). During the study period, 177 (7%) virtual ward admissions resulted in a hospital (re)admission and 9 (<1%) died. First NEWS2 = 0−2 in 1,651/2,479 (67%) admissions, NEWS2 = 3−4 in 569 (23%), NEWS2 = 5−6 in 204 (8%) and NEWS = 7+ in 55 (2%) admissions. Maximum NEWS2

**Data availability statement:** Data may be obtained from a third party and are not publicly available. The data used in the study are collected by Sirona Care & Health as part of their routine care and support. Sharing of pseudo-anonymised data with the University of Bristol was underpinned by a data sharing agreement and ethical approval which solely covers the purposes of this study. This study included secondary data analysis without patient consent, so it is not appropriate to share the data, even if it is anonymised. We obtained a demographics dataset and a clinical observations dataset, as detailed in the methods. We did not obtain any special privileges to obtain this data, but we did fund a data analyst to extract it. Data requests can be made directly to Sirona Care & Health (sirona.hello@nhs.net). Our analysis code is publicly available on our GitHub repository (https://github.com/ARCWest-ADS/NEWS_AND_VIRTUAL_WARDS).

**Funding:** This project was funded by NHS England South West and supported by the National Institute for Health and Care Research Applied Research Collaboration West (NIHR ARC West). The views expressed in this article are those of the authors and not necessarily those of NHS England, NHS Improvement, the NIHR or the Department of Health and Social Care.

**Competing interests:** The authors have declared that no competing interests exist.

was reasonably low during most admissions, with 800/2,479 (32%) NEWS2 = 0−2, 887 (36%) NEWS2 = 3−4, 568 (23%) NEWS2 = 5−6, and 224 (9%) NEWS2 = 7 +. The clinical acuity of most patients remained stable, with 964/2,331 (41%) deteriorating by 1−2 points and 696 (30%) not deteriorating at all. Both first NEWS2 and maximum NEWS2 had poor ability to predict hospital (re)admission (AUC 0.55 [95% CI 0.51–0.60] and 0.55 [95% CI 0.50–0.59], respectively).

### Conclusion

Most remotely monitored patients had low clinical acuity on admission to virtual wards, however 10% had high clinical acuity with a NEWS2 value of at least 5. The distribution of NEWS2 on admission to virtual wards was very similar to the distribution of NEWS2 on admission to acute hospitals, as identified in the 2022 Society for Acute Medicine Benchmarking Audit. NEWS2 had poor predictive accuracy in this setting. However, hospital (re)admission rates were low (7%) so this should be interpreted with caution.

## Introduction

Virtual wards are an NHS priority [1–4], designed to deliver care to people who have become acutely unwell and would otherwise be in hospital, or as a way of supporting earlier discharge following an acute admission [5]. They enable the provision of acute care, monitoring and treatment in peoples preferred place of care, such as in their own home or a care home.

In Bristol, North Somerset and South Gloucestershire (BNSSG), many of the virtual wards within the NHS@home service use digital technology to help remotely monitor patients and are supported to do so by Sirona Care & Health [6] (the care provider) and Doccla [7] (the software provider). Patients are provided with equipment to measure their clinical observations at home [8]. They are asked to use the equipment at individually determined times throughout the day (usually up to 3 times per day) and input their observations into a smartphone/tablet which uploads them onto the Doccla Dashboard. The digitally recorded clinical observations include respiratory rate, oxygen saturations, systolic blood pressure, pulse rate, and temperature. The clinical team are alerted by the system when a patients vital signs fall outside of agreed parameters [9]. Virtual wards which use this technology in BNSSG look after patients with respiratory illness, heart failure, frailty, and outpatient parenteral antimicrobial therapy (OPAT), as well as other acute medical or surgical presentations (group together as 'general') [10]. As well as remote monitoring of physiological observations, patients on these wards may receive in-person clinical care including medical assessment, taking blood, urine and sputum specimens, dressing changes, and delivering of intravenous medications including diuretics and antibiotics, depending on their clinical need. Many patients are visited regularly, while others are primarily monitored remotely and may only be seen face-to-face if their observations or pathway symptoms specific

questionnaire completion suggest intervention might be required. This paper focuses on remotely monitored patients, whether or not they also had face-to-face visits.

Quantitative evaluations of the virtual ward model to date have largely focused on patient demographics, length of stay, the financial impact, the best delivery method, and workforce [11–14]. Many papers focus on the use of virtual wards during the COVID-19 pandemic [15], or specifically on hospital care delivered at home, but none have focused on how sick these patients are (their clinical acuity) [16]. Although the clinicians who admit patients to a virtual ward will know their individual observations and resultant clinical acuity at the point of admission, little is known about the collective clinical acuity of patients admitted onto virtual wards, how this changes throughout the period they are under virtual ward care (physiological stability), and their subsequent outcomes including hospital (re)admissions and virtual ward mortality.

Clinical observations such as respiration rate and oxygen saturation, amongst others, may provide an indication of the clinical acuity of a patient. Further, the National Early Warning Score (NEWS2) is a scoring system that combines a number of such clinical observations (respiratory rate, oxygen saturation, systolic blood pressure, pulse rate, temperature, supplemental oxygen and level of consciousness) to calculate an overall score which provides an overall indicator of clinical acuity. NEWS was developed by the Royal College of Physicians (RCP) in 2012 [17] and was updated in 2017 to become NEWS2. It is mandated by NHS England for use in the ambulance service and acute hospitals [18], and is also recommended for use across the whole NHS as a common language [18]. National Institute for Health and Care Excellence (NICE) guidance NG51 2024 [19] also recommends the use of NEWS2 to support identifiaction of patients with possible sepsis. NHS England recommends that NEWS2 should be used as part of the initial assessment when patients are admitted to Virtual Wards [9]. While clinical observations alone cannot provide the full picture of clinical acuity, NEWS2 is a strong indicator of acute clinical illness and deterioration [20], and can be monitored remotely, and as such is used to represent clinical acuity in this paper.

The aim of this study was to understand the clinical acuity (measured using NEWS2) of people admitted to virtual wards in BNSSG who submitted remote monitoring data to the Doccla platform under the care of Sirona Care & Health.

The objectives were to: extract data from digital healthcare records relating to the digitally recorded clinical observations, care pathways, outcomes, and demographics of people admitted to virtual wards; use the digitally recorded clinical observations to calculate NEWS2; understand the clinical acuity of patients on different care pathways; and assess the association between NEWS2 and hospital (re)admission and virtual ward mortality.

## Methods

### Design and setting

This was a retrospective observational cohort study of remotely monitored patients on virtual wards in BNSSG Integrated Care System (ICS). It included all patients who were admitted to a digitally-enabled ward between October 2023 and February 2025. Only clinical observations recorded using the Doccla equipment and software are included in this study; this was provided to approximately 60% of patients who are on these virtual wards (patients who had regular visits from a clinician and/or were felt to be unable to use the technology required were not provided with Doccla kit- these patients were therefore predominantly monitored in person rather than remotely). Data extraction occurred on 4th June 2025.

Patients were eligible if they met the above criteria and were aged 16 + years on admission. No formal sample size was calculated as all patients who met the eligibility criteria were included.

### Data acquisition

Pseudo-anonymised data were obtained from Sirona Care & Health, the lead provider of virtual wards in BNSSG. They collate clinical data collected on virtual wards using the Doccla system, alongside patients' primary care records (EMIS). Ethical approval was obtained through the University of Bristol Faculty of Health Sciences Research Ethics Committee

(FREC; ID 15007). All analysed data were routinely collected and pseudo-anonymised prior to transfer to the research team so patient consent was not required.

Data included patient age, sex, care pathway (e.g., frailty or respiratory), source of admission (e.g., acute hospitals, the ambulance service, or GPs), virtual ward admission date, virtual ward discharge date, and the outcome of their virtual ward care (i.e., virtual ward care complete, patient (re)admitted to hospital, or patient died). All NEWS2 component clinical observations (respiratory rate, oxygen saturations, systolic blood pressure, pulse rate, level of consciousness, temperature, oxygen requirement, and hypercapnic respiratory failure status), and Doccla calculated NEWS2 values, along with date and time of each set of observations, were also requested.

Patients who were admitted in error, declined invitation or self-discharged, could not be contacted, moved out-of-area before discharge, or who were still in-patients at the point of data extraction (all identified in the outcome field), were excluded from the analysis. All discrete episodes of virtual ward care were included. Virtual ward care during the same period across multiple care pathways was counted as one admission (although sometimes a patient may be in multiple virtual wards at the same time), and patients who had multiple separate admissions were included multiple times (the number of times this occurs is described).

### Data derivations

Age was categorised into six groups: 16–39, 40–49, 50–59, 60–69, 70–79, and 80 + years. Virtual ward length of stay was calculated as the number of days between admission date and discharge date (i.e., the date they were discharge from the virtual ward, (re)admitted to hospital, or died).

Clinical observations (e.g., respiratory rate, oxygen saturation, etc) were graphically explored and clinically unlikely/implausible values were removed, following discussion with clinical co-authors. Included ranges were: 3–60 breaths per minute for respiratory rate, 50–100% for oxygen saturation, 60–260 mmHg for systolic blood pressure, 20–181 beats per minute for pulse rate, and 30-41°C for temperature. The first recording of each clinical observation following admission to the virtual ward was identified for each admission. Further, for each admission, minimum and maximum values of each clinical observation across the episode of virtual ward stay were calculated.

NEWS2 (scored 0–20, with higher scores indicating higher clinical acuity) and NEWS2 component scores (e.g., respiratory rate 0–3) for each clinical observation were derived following the rules on the NEWS2 scoring card [21]. Oxygen requirement, level of consciousness, and hypercapnic respiratory failure status were collected on the Doccla platform but not shared with Sirona. Therefore, for our derivation of NEWS2, we assumed patients were alert and not newly confused, and that there was no additional oxygen therapy required. Further, oxygen saturation component score derivations were made using scale 1 [22]. The presented NEWS2 scores are therefore more analogous to the original NEWS, however, in line with current clinical practice we have referred to them as NEWS2 throughout this manuscript.

Three NEWS2 values were considered for analysis: The 'first' NEWS2 for each admission (based on the first full set of digitally recorded observations); the maximum NEWS2 for each admission during the virtual ward episode (which in some cases was the same as the first score); and a change score, calculated as the maximum score minus the first score. The change score was recorded as missing for admissions where only one derived NEWS2 value could be calculated. First and maximum component scores were derived in the same way; change scores were not calculated for component scores.

NEWS2 values were grouped into five categories for analysis: 0, 1–2, 3–4, 5–6,and 7 + , in line with hospital escalation trigger scores of 3, 5 and 7 [23]. Similarly, change scores were also grouped into five categories: did not deteriorate, and deteriorated by 1–2, 3–4, 5–6, and 7 + .

Doccla calculated NEWS2 values were also available for some sets of observations (and did not suffer from the missing NEWS2 components like the derived NEWS2 did). As these were available for far less patients, we chose to focus our paper on the derived NEWS2 values; however, we present the Doccla calculated NEWS2 values as a sensitivity analysis (see below).

## Statistical analysis

Continuous data were summarised using means & standard deviations, or medians & interquartile ranges (IQR) if the data were skewed. For clinical observations, ranges are also presented. Categorical data were summarised using counts & percentages. Counts and percentages of first and maximum NEWS2 values and their component parts are presented graphically.

The outcomes of interest were virtual ward mortality and hospital admission from the virtual ward; as many (but not all) patients were transferred to a virtual ward following a previous hospital stay, and therefore admission to hospital would be a readmission for them, we have used the term 'hospital (re)admission' throughout the paper to describe hospital admissions following virtual ward stays. Given the very small number of admissions which end in mortality during virtual ward care, we assessed the sensitivity and specificity of first and maximum NEWS2 to predict hospital (re)admission, and only present virtual ward mortality descriptively. Receiver operating characteristic curves (sensitivity against 1-specificity) were constructed and area under the curve (AUC) calculated along with 95% CIs. AUC values of 0.70–0.79 were considered acceptable, 0.80–0.89 excellent and ≥0.90 outstanding [24].

## Sensitivity analyses

We performed sensitivity analyses to assess (1) the AUC for first and maximum NEWS2 in predicting hospital (re)admission excluding those on the respiratory pathway (as these patients are most likely to be affected by the missing oxygen saturation scale information),and (2) the AUC for first and maximum Doccla calculated NEWS2 in predicting hospital (re)admission.

All analyses were based on complete cases, and no adjustments were made for missing data, however, missing data are described. All data checking, cleaning, and analyses were conducted in Stata 19.5.

## Results

### Demographics and care pathways

Between 01 October 2023 and 28 February 2025, there were 4,279 eligible admissions onto digitally-enabled virtual wards in BNSSG. Of these, 2,533 (59%) has at least one observation recorded on the Doccla system and were included in our study population. Patients who were younger, admitted from hospital, and on respiratory, OPAT and heart failure pathways were more likely to have clinical observations, and those who were older and on frailty or general pathways were less likely to have clinical observations (S1 Table in S1 Appendix).

The 2,533 admissions with observations were in 2,207 patients; 1,856 patients (84%) had one admission, 256 (12%) had two admissions, 95 (4%) had three or more admissions. The analysis is at the admission level rather than the person level.

There were 1,040 (41%) admissions to a respiratory pathway, 446 (18%) to a frailty pathway, 483 (19%) OPAT, 178 (7%) heart failure, and 453 (18%) to a general (non-specific) pathway (Table 1). The median age on admission was 73 years (IQR 60–82) and 1,391 (55%) were female. The median length of virtual ward stay was 10 days (IQR 6–14). The majority of patients (2,202; 87%) were admitted from an acute hospital. Most patients (2,347; 93%) completed their virtual ward care and were discharged, 177 (7%) were (re)admitted to hospital, and 9 (<1%) died while on a virtual ward (Table 1).

### Clinical acuity

In 2,265/2,533 (89%) admissions, first digital observations were taken within 2 days of admission to the virtual ward. For the first set of observations, the median respiratory rate was 17 per minute (IQR 15–20, range 8–60), median oxygen saturation was 96% (IQR 94–97, range 64–100), median systolic blood pressure was 127 mmHg (IQR 113–144, range 60–260), median pulse rate was 81 beats per minute (IQR 71–90, range 37–157), and median temperature was 36.5°C (IQR 36.3 to 36.8, range 33.3 to 39.9; Table 2). Minimum and maximum values are also presented in Table 2.

**Table 1. Patient demographics.**

| | All virtual ward admissions (n = 2,533) | |
|---|---|---|
| | n | % |
| **Age (median, IQR):** | 73 | (60, 82) |
| 16-39 years | 184 | 7.3% |
| 40-49 years | 164 | 6.5% |
| 50-59 years | 280 | 11.1% |
| 60-69 years | 432 | 17.1% |
| 70-79 years | 677 | 26.7% |
| 80+ years | 796 | 31.4% |
| **Sex:** | | |
| Male | 1,142 | 45.1% |
| Female | 1,391 | 54.9% |
| **Source of referral:** | | |
| Acute Hospital | 2,202 | 86.9% |
| Ambulance Service | 34 | 1.3% |
| Community Health Service | 176 | 6.9% |
| Emergency Care Department | 1 | 0.0% |
| GP Practice | 119 | 4.7% |
| Not known | 1 | 0.0% |
| **Pathway*:** | | |
| Respiratory | 1,040 | 41.1% |
| Frailty | 446 | 17.6% |
| OPAT | 483 | 19.1% |
| Heart Failure | 178 | 7.0% |
| General | 453 | 17.9% |
| **Outcome of care:** | | |
| Virtual ward care complete | 2,347 | 92.7% |
| (Re)admitted to hospital | 177 | 7.0% |
| Patient died | 9 | 0.4% |
| **Virtual ward length of stay (days; median, IQR)** | 10 | (6, 14) |

*Patients may be on more than one pathway during the same admission.

IQR=Interquartile range. OPAT=Outpatient Parenteral Antimicrobial Therapy.

NEWS2 values could be calculated in 2,479 (98%) admissions. First NEWS2 was 0 in 600/2,479 patients (24%), NEWS2 = 1–2 in 1,051 patients (42%), NEWS2 = 3–4 in 569 (23%), NEWS2 = 5–6 in 204 (8%) and NEWS2 = 7+ in 55 patients (2%; Fig 1 and S2 Table in S1 Appendix). Maximum NEWS2 was reasonably low in the majority of patients, with 800/2,479 (32%) NEWS2 = 0–2, 887 (36%) NEWS2 = 3–4, 568 (23%) NEWS2 = 5–6, and 224 (9%) NEWS2 = 7+ (Fig 1 and S2 Table in S1 Appendix). Most patients had little or no deterioration (964/2,331 [41%] deteriorated by 1–2 NEWS2 points and 696 [30%] did not deteriorate), however 17 (1%) patients' NEWS2 value deteriorated by 7+ (Fig 1 and S2 Table S1 Appendix).

Most patients scored zero on each of the component scores for their first score, with the exception of oxygen saturation where 46% scored at least one (Fig 2 and S2 Table in S1 Appendix). For the maximum component scores, many patients scored at least one across all of the measurements; the maximum oxygen saturation scores were particularly high (Fig 2 and S2 Table in S1 Appendix).

**Table 2. First, minimum and maximum clinical observations.**

| | All virtual ward admissions (n = 2,533) | | |
|---|---|---|---|
| | **Median** | **IQR** | **Range** |
| **First:** | | | |
| Respiratory rate (per minute)[1] | 17 | (15, 20) | (8, 60) |
| Oxygen saturation (%)[2] | 96 | (94, 97) | (64, 100) |
| Systolic blood pressure (mmHg)[3] | 127 | (113, 144) | (60, 260) |
| Pulse rate (per minute)[4] | 81 | (71, 90) | (37, 157) |
| Temperature (°C)[5] | 36.5 | (36.3, 36.8) | (33.3, 39.9) |
| **Minimum:** | | | |
| Respiratory rate (per minute)[1] | 13 | (12, 15) | (3, 59) |
| Oxygen saturation (%)[2] | 94 | (91, 96) | (50, 100) |
| Systolic blood pressure (mmHg)[3] | 110 | (101, 121) | (60, 186) |
| Pulse rate (per minute)[4] | 69 | (61, 77) | (21, 157) |
| Temperature (°C)[5] | 36.1 | (35.8, 36.4) | (30.0, 39.2) |
| **Maximum:** | | | |
| Respiratory rate (per minute)[1] | 21 | (19, 24) | (10, 60) |
| Oxygen saturation (%)[2] | 98 | (96, 99) | (77, 100) |
| Systolic blood pressure (mmHg)[3] | 146 | (130, 161) | (92, 260) |
| Pulse rate (per minute)[4] | 92 | (82, 102) | (44, 181) |
| Temperature (°C)[5] | 36.8 | (36.6, 37.2) | (33.3, 40.0) |

IQR = Interquartile range. [1]Missing data for 22 patients. [2]Missing data for 5 patients. [3]Missing data for 22 patients. [4]Missing data for 12 patients. [5]Missing data for 17 patients.

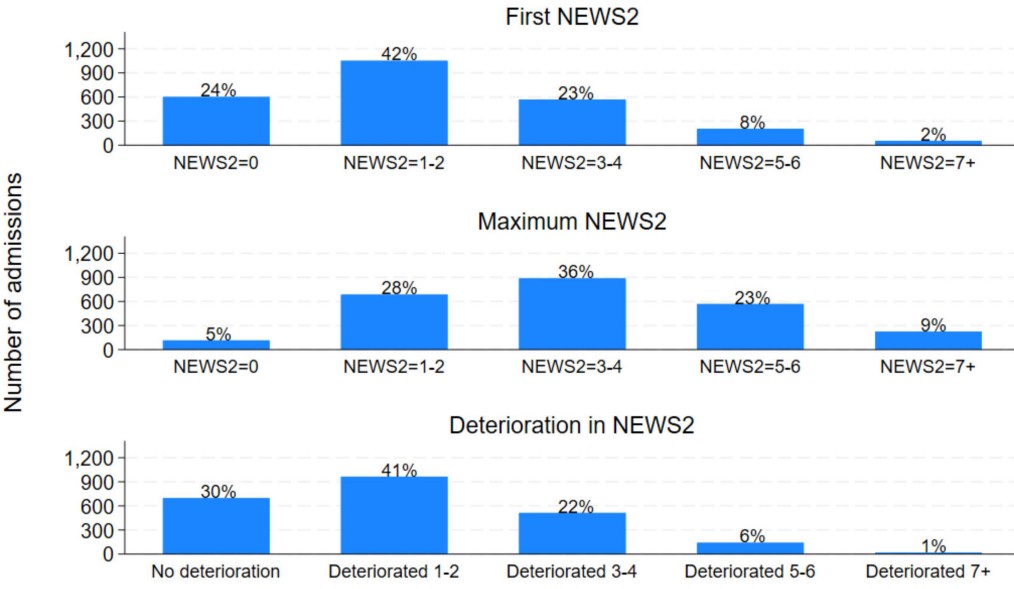

**Fig 1. Distribution of first, maximum and deterioration in NEWS2 values.**

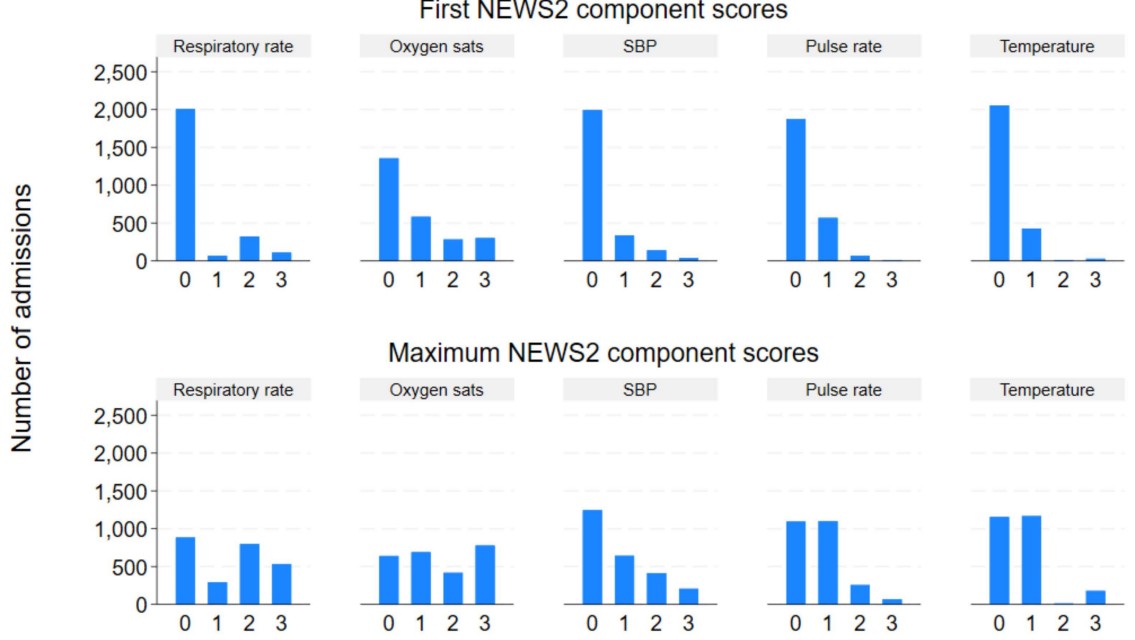

**Fig 2. Distributions of first and maximum component NEWS2 values.** SBP = Systolic Blood pressure.

### Using NEWS2 to predict hospital (re)admission and virtual ward mortality

Only 177/2,533 patients (7%) were (re)admitted to hospital, of whom 166 also had NEWS2 data. Patients with low first NEWS2 were slightly less likely to be (re)admitted to hospital than those with higher first NEWS2 (e.g., 31/600 [5.1%] patients with first NEW2 = 0–2, 44/569 [7.7%] with first NEWS = 3–4, and 7/55 [12.7%] with first NEWS = 7 + were (re)admitted); this pattern was less clear for maximum NEWS2 (S3 Table in S1 Appendix). According to AUC, first and maximum NEWS2 had little predictive ability for hospital (re)admission (AUC 0.55 [95% CI 0.51–0.60] and 0.55 [95% CI 0.50–0.59], respectively; Figs 3 and 4). Similarly, when excluding patients on the respiratory pathway (sensitivity analysis 1), the ability of first and maximum NEWS2 to predict hospital (re)admission remained poor (AUC 0.55 [95% CI 0.50–0.60] and 0.57 [95% CI 0.51–0.62], respectively). Hospital (re)admissions by patient demographics are presented as supplemental material (S3 Table in S1 Appendix). Numbers were too low to predict AUC for mortality (9/2,533; < 1%) but of the 8 patients who died and also had a NEWS2 value calculated, first NEWS2 = 0 in 1 patients, NEWS2 = 1–2 in 3 patients and NEWS2 = 3–4 in 4 patients; maximum NEWS2 = 1–2 in 2 patients, NEWS2 = 3–4 in 4 patients, NEWS2 = 5–6 in 1 patients and NEWS2 = 7+ in 1 patient.

### Clinical acuity by demographics and care pathways

There were no substantial differences in NEWS2 values between patients of different demographics (S1-S2 Figs in S1 Appendix). Patients on the respiratory, frailty and heart failure pathways tended to have slightly higher first NEWS2 and maximum NEWS2 values than patients on OPAT or general pathways, and patients on the frailty pathway were slightly less likely to have deteriorating NEWS2 values (S3 Fig in S1 Appendix).

### Comparison with Doccla calculated NEWS2 values (sensitivity analysis 2)

There were 982 admissions (39%) that had at least one NEWS2 value calculated by the Doccla system. For sets of observations where Doccla NEWS2 values were calculated and we were able to calculate NEWS2, 81% of values were

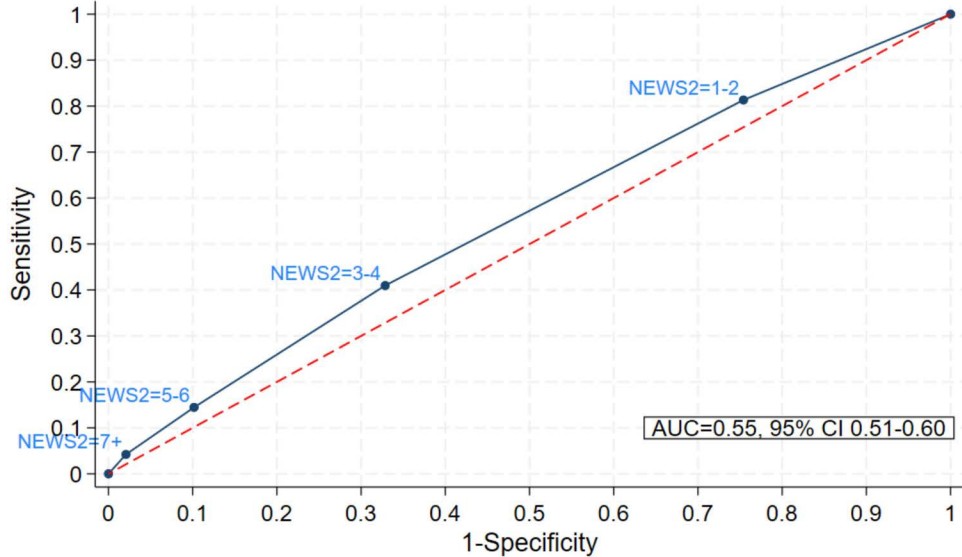

**Fig 3. Sensitivity and specificity of first NEWS2 to predict hospital (re)admission.**

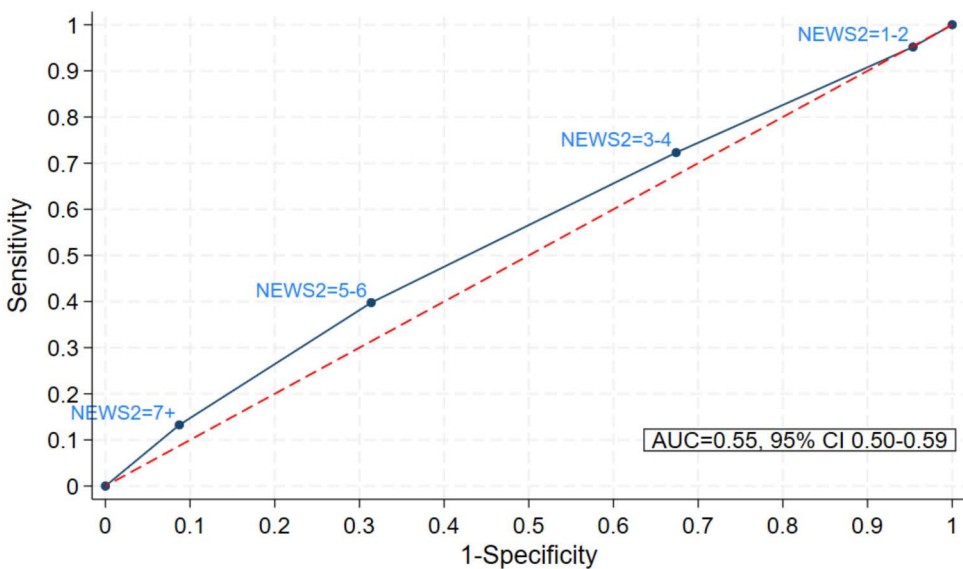

**Fig 4. Sensitivity and specificity of maximum NEWS2 to predict hospital (re)admission.**

concordant. As with derived NEWS2, neither first or maximum Doccla calculated NEWS2 had any predictive ability for hospital (re)admissions (AUC 0.54 [95% CI 0.47–0.62] and 0.52 [95% CI 0.44–0.59], respectively.

## Discussion

### Summary of results

This study included 2,533 remotely monitored patients admitted to virtual wards in BNSSG across five care pathways, with 41% on the respiratory pathway, 19% on the OPAT pathway, 18% frailty, 18% general and 7% on the heart failure

pathway.. 87% were referred into virtual wards from acute hospitals. The median length of virtual ward stay was 10 days (IQR 6–14). Only 7% of patients were (re)admitted from a virtual ward into hospital and <1% died. Many patients on virtual wards were 70+years (58%), but 7% were 16–39 years old. Most remotely monitored virtual ward patients had low clinical acuity on admission (NEWS2 = 0–2 for 67% of patients). Further, most patients were physiologically stable, with 71% deteriorating by ≤2 NEWS2 points during their admission. Interestingly, despite their good predictive value in other settings, prediction of (re)admission to hospital using NEWS2 in this setting was poor.

## Strengths and limitations

This is the first paper to date that focuses on the clinical acuity of remotely monitored patients on virtual wards, and should therefore be used to inform Integrated Care Boards (ICBs) about the clinical acuity of patients and use of NEWS2 on virtual wards [1,2]. The main strength of this paper is the inclusion of a large number of patients, across differing geographic regions in BNSSG and several different patient pathways. This makes our findings more generalisable to other areas of the UK than if we had just focused on one pathway, which is a more common approach in this type of research. However, a limitation of the study is that it only focuses on remotely monitored virtual ward patients, rather than all patients admitted to these wards. Kit is prioritised for patients who are best placed to use it, this means that many of the more elderly patients, many of whom are on the frailty pathway, do not have digital observations recorded on this system.

The results of this paper have highlighted the lack of predictive ability of NEWS2 for hospital (re)admission for remotely monitored patients on virtual wards, demonstrating opportunities for further development and optimisation of digitally enabled pathways in BNSSG. However, along with potential power issues due to small numbers of (re)admissions, importantly, the measures of clinical acuity reported may not be representative of all the patients on these wards for a number of reasons. Firstly, as mentioned above, only 59% of patients on these wards had their clinical observations digitally recorded (and therefore remotely monitored), and for 11% of remotely monitored admissions, their first digital observations were more than 2 days after virtual ward admission. Secondly, some components of the NEWS2 scoring system (oxygen saturation scale, supplemental oxygen and level of consciousness) were not shared with Sirona and therefore we were unable to calculate accurate NEWS2 values. This issue will mainly affect patients on the respiratory pathway, many of whom would be on scale two for their oxygen saturation; by assuming they are on scale one we would have scored them higher for this component (and therefore overall NEWS2 value) than we should. However, when compared to the observations where NEWS2 values were also recorded in Doccla, 81% of values were concordant. Further, a sensitivity analysis of the AUC for hospital (re)admission revealed very little difference in predictive ability when excluding patients on the respiratory pathway, or when using the Doccla calculated NEWS2 values instead. We have highlighted the data issues we have found to the care providers who have developed these services to help better link up communication between the systems (and therefore clinicians). Finally, we identified some implausible values in the clinical observations (271/123,486 observations [0.2%]) which we chose to exclude for the purpose of analysis. Data entry errors of this magnitude are expected in this type of data collection. In terms of the actual service, recording of these extremely high or low values would trigger a phone call to the patient from the clinical team to check how they were feeling, alongside a review of symptoms and repetition of the observations whilst on the telephone. The error would be promptly identified, and no further action would be needed.

## Comparison with other literature

The use of NEWS2 on admission to virtual wards enables teams to look at the spread of scores and provide some insight into how sick patients are. The distribution of first NEWS2 values observed (67% NEWS2 = 0–2, 23% of NEWS2 = 3–4, 7% NEWS2 = 5–6, and 2% of NEWS2 = 7+) was very similar to that reported by the Society for Acute Medicine Benchmarking Audit (SAMBA) for patients admitted to acute medical wards (71% of unplanned attendances

had NEWS2 = 0–2, 15% NEWS2 = 3–4, 7% NEWS2 = 5–6, 7% NEWS2 = 7+) [25]. This is contrary to the wider perception that people looked after on virtual wards are of a lower acuity than patients admitted to hospital and are therefore not patients who would otherwise be in hospital, but patients for whom a safety-netting service is being provided. In fact, given that we have only included patients who are remotely monitored, and we know these patients are younger and less likely to be admitted to hospital/ die than the full BNSSG virtual ward population (S1 Table in S1 Appendix), it is likely that if we were able to include all virtual ward patients, they would have a higher clinical acuity than those measured in SAMBA. The SAMBA report was only a 24-hour snapshot so could not provide details of the stability of patients; it is possible that virtual ward patients are more physiologically stable than long-term hospital patients and this is why virtual wards are appropriate for them. NEWS2 distributions were also similar to those reported in a recent evaluation of hospital at homes services in Oxford, Buckinghamshire and Berkshire West [26]. However, the predicative accuracy of NEWS2 values to predict patient deterioration (hospital (re)admission) in this setting was poor. One reason could be that virtual ward patients present at hospital when they are feeling acutely unwell, rather than taking additional observations themselves, so their potential increase in NEWS2 value is not being recorded on the remote monitoring system. Although NEWS2 has been recommended for use in virtual wards, most evaluations have not described clinical acuity [27]. Instead, they have focussed on areas such as patient demographics, patient outcomes, patient and staff experience, cost savings and healthcare utilisation [11,13,28–30]. The reporting of vital signs, where included, was part of monitoring a patient and supporting escalation decisions [11,29], rather than reporting ranges of NEWS2 values or clinical acuity. One of the challenges of this work was collecting all the vital signs needed to calculate NEWS2. In particular, respiratory rate is often poorly recorded, despite its clinical value [31,32]. This was a particular issue in the first 9 months in BNSSG virtual wards (January-September 2023), and as such we chose to exclude patients admitted during that period. One of the main issues of using digital technology is that of digital exclusion due to a lack of understanding and/or internet connection; this is reflected in this data with only 59% of patients having observations recorded using Doccla, and far fewer of those on the frailty pathway having observations. A recent evaluation of the practicalities of using digital technology on virtual wards explored factors such as patients recording vital signs in more detail and found similar difficulties [33]. We have addressed this limitation by explicitly detailing the inclusion criteria as virtual ward admissions which are remotely monitored, rather than the whole of the BNSSG virtual ward population.

### Implications for research and/or practice

The distribution of NEWS2 values of remotely monitored patients admitted to virtual wards is similar to that of patients admitted to acute medical wards [25]. Hospital (re)admission rates and mortality following virtual ward care are low, which should provide reassurance that care outside of hospital is appropriate for these patients. However, the poor predictive value of NEWS2 for (re)hospitalisation in this setting should be further investigated, ideally in a large national evaluation. Given the recommendations in the Darzi review [3] and the 2025 NHS 10 year plan [4] to expand virtual ward services, it is important to further investigate the clinical acuity of patients on these wards as well as how best to integrate digital technologies given the presented complexities.

### Conclusion

Most remotely monitored patients had low clinical acuity on admission to virtual wards, however 10% had a NEWS2 value of at least 5. The distribution of NEWS2 on admission to virtual wards was very similar to the distribution of NEWS2 on admission to acute hospitals, as identified in the 2022 Society for Acute Medicine Benchmarking Audit. NEWS2 had poor predictive accuracy for predicting hospital (re)admission in remotely monitored virtual ward admissions in BNSSG. However, hospital (re)admission rates were low (7%) so this should be interpreted with caution.

## Supporting information

**S1 Appendix. Supplemental Tables and Figures.**
(DOCX)

## Author contributions

**Conceptualization:** Lauren J Scott, Rebecca Winterborn, Jenny Tomkinson, Alison Tavaré.

**Data curation:** Lauren J Scott.

**Formal analysis:** Lauren J Scott.

**Funding acquisition:** Lauren J Scott, Joanne Appleton, Alison Tavaré.

**Methodology:** Lauren J Scott.

**Project administration:** Lauren J Scott, Chris Penfold.

**Supervision:** Isobel Ward, Frank de Vocht.

**Validation:** Rebecca Winterborn.

**Visualization:** Lauren J Scott.

**Writing – original draft:** Lauren J Scott, Rebecca Winterborn, Alison Tavaré.

**Writing – review & editing:** Joanne Appleton, Chris Penfold, Jenny Tomkinson, Isobel Ward, Frank de Vocht.

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
