## [Decision Letter · Decision Letter 0]

21 Oct 2025

Dear Dr. Scott,

**In spite of the major concerns that have ben raised, the authors have been granted a second chance to improve the manuscript. please address all the points that have been raised**

plosone@plos.org. . . . A rebuttal letter that responds to each point raised by the academic editor and reviewer(s). You should upload this letter as a separate file labeled 'Response to Reviewers'.A marked-up copy of your manuscript that highlights changes made to the original version. You should upload this as a separate file labeled 'Revised Manuscript with Track Changes'.An unmarked version of your revised paper without tracked changes. You should upload this as a separate file labeled 'Manuscript'.

We look forward to receiving your revised manuscript.

Kind regards,

Tarek Samy Abdelaziz, MD,FRCP

Academic Editor

PLOS ONE

**Journal Requirements:**

1. When submitting your revision, we need you to address these additional requirements. Please ensure that your manuscript meets PLOS ONE's style requirements, including those for file naming. The PLOS ONE style templates can be found at https://journals.plos.org/plosone/s/file?id=wjVg/PLOSOne_formatting_sample_main_body.pdf and https://journals.plos.org/plosone/s/file?id=ba62/PLOSOne_formatting_sample_title_authors_affiliations.pdf 2. We note that the grant information you provided in the ‘Funding Information’ and ‘Financial Disclosure’ sections do not match.  When you resubmit, please ensure that you provide the correct grant numbers for the awards you received for your study in the ‘Funding Information’ section. 3. Thank you for stating in your Funding Statement: This project was funded by NHS England South West and supported by the National Institute for Health and Care Research Applied Research Collaboration West (NIHR ARC West). The views expressed in this article are those of the authors and not necessarily those of NHS England, NHS Improvement, the NIHR or the Department of Health and Social Care.  Please provide an amended statement that declares *all* the funding or sources of support (whether external or internal to your organization) received during this study, as detailed online in our guide for authors at http://journals.plos.org/plosone/s/submit-now.  Please also include the statement “There was no additional external funding received for this study.” in your updated Funding Statement. Please include your amended Funding Statement within your cover letter. We will change the online submission form on your behalf. 4. We note that you have indicated that there are restrictions to data sharing for this study. For studies involving human research participant data or other sensitive data, we encourage authors to share de-identified or anonymized data. However, when data cannot be publicly shared for ethical reasons, we allow authors to make their data sets available upon request. For information on unacceptable data access restrictions, please see http://journals.plos.org/plosone/s/data-availability#loc-unacceptable-data-access-restrictions.  Before we proceed with your manuscript, please address the following prompts: a) If there are ethical or legal restrictions on sharing a de-identified data set, please explain them in detail (e.g., data contain potentially identifying or sensitive patient information, data are owned by a third-party organization, etc.) and who has imposed them (e.g., a Research Ethics Committee or Institutional Review Board, etc.). Please also provide contact information for a data access committee, ethics committee, or other institutional body to which data requests may be sent. b) If there are no restrictions, please upload the minimal anonymized data set necessary to replicate your study findings to a stable, public repository and provide us with the relevant URLs, DOIs, or accession numbers. Please see http://www.bmj.com/content/340/bmj.c181.long for guidelines on how to de-identify and prepare clinical data for publication. For a list of recommended repositories, please see https://journals.plos.org/plosone/s/recommended-repositories. You also have the option of uploading the data as Supporting Information files, but we would recommend depositing data directly to a data repository if possible. Please update your Data Availability statement in the submission form accordingly. 5. For studies involving third-party data, we encourage authors to share any data specific to their analyses that they can legally distribute. PLOS recognizes, however, that authors may be using third-party data they do not have the rights to share. When third-party data cannot be publicly shared, authors must provide all information necessary for interested researchers to apply to gain access to the data. (https://journals.plos.org/plosone/s/data-availability#loc-acceptable-data-access-restrictions)  For any third-party data that the authors cannot legally distribute, they should include the following information in their Data Availability Statement upon submission:a) A description of the data set and the third-party sourceb) If applicable, verification of permission to use the data setc) Confirmation of whether the authors received any special privileges in accessing the data that other researchers would not haved) All necessary contact information others would need to apply to gain access to the data 6. Your ethics statement should only appear in the Methods section of your manuscript. If your ethics statement is written in any section besides the Methods, please move it to the Methods section and delete it from any other section. Please ensure that your ethics statement is included in your manuscript, as the ethics statement entered into the online submission form will not be published alongside your manuscript. 7. We note that you have referenced (Lyndon H, Viney T.) which has currently not yet been accepted for publication. Please remove this from your References and amend this to state in the body of your manuscript: (“Lyndon H, Viney T. [Submitted]”) as detailed online in our guide for authors http://journals.plos.org/plosone/s/submission-guidelines#loc-reference-style 8. Please include captions for your Supporting Information files at the end of your manuscript, and update any in-text citations to match accordingly. Please see our Supporting Information guidelines for more information: http://journals.plos.org/plosone/s/supporting-information. 9. If the reviewer comments include a recommendation to cite specific previously published works, please review and evaluate these publications to determine whether they are relevant and should be cited. There is no requirement to cite these works unless the editor has indicated otherwise. 

**Additional Editor Comments:**

The reviewers have raised major concerns about data accuracy and comparison to between virtual ward and acute wards

It has been decided that the authors will be granted a second chance to improve the manuscript provided that the authors will address those major concerns

Reviewers' comments:

**Comments to the Author**

1. Is the manuscript technically sound, and do the data support the conclusions?

Reviewer #1: Partly

2. Has the statistical analysis been performed appropriately and rigorously?

Reviewer #1: N/A

3. Have the authors made all data underlying the findings in their manuscript fully available?

Reviewer #1: Yes

4. Is the manuscript presented in an intelligible fashion and written in standard English?

Reviewer #1: Yes

**Reviewer #1:** The authors conduct a retrospective analysis of physiological parameters for patients managed on a virtual ward network. The authors conduct a retrospective analysis of physiological parameters for patients managed on a virtual ward network. The authors conduct a retrospective analysis of physiological parameters for patients managed on a virtual ward network. The authors conduct a retrospective analysis of physiological parameters for patients managed on a virtual ward network.

There are a number of areas of major concerns which need to be considered.

The first area is the definition and description of the hospital level interventions delivered on this virtual ward network. This is missing and it is essential for interpretation of the manuscript. Was their point of care ultrasound delivered diagnostics, point of care blood monitoring, intravenous therapies etc.. Delivering physiological monitoring in isolation is insufficient to meet this criteria.

Further why have OPAT services been included? OPAT services has been in existence since the last 1990s and should not be regarded as meeting the criteria for virtual ward admission unless further interventions are also being delivered in the service.

The second area that needs consideration is the equivalence in the paper given to clinical acuity and physiological stability (as measured by NEWS2). It does not appear that any other markers of clinical acuity are examined in the paper and by themselves physiological parameters are not a good measure of clinical acuity.

The third area is the NEWS2 data itself:-

1. Only 57% of patients have a NEWS2 score that can be calculated. This is an almost insurmountable issue and the conclusions made in the paper should be stated with significantly less confidence in the context of this missing data.

2. How did the authors derive their implausible physiological parameters to be excluded? How many of these were there? This raises concerns about the accuracy of the device used. How many patients had a respiratory rate of >50 for example. Further, when do the authors think people may have a temperature >43?

3. "Clinical acuity" cannot be compared to acute hospital care when 12% of initial observations were taken >48 hours after "admission" to the virtual ward.

The fourth area is comparison to hospital care:-

1. Over 80% of the patients have undergone a period of acute inpatient care, assessment and treatment prior to discharge to a "virtual ward". Therefore, this is an entirely different cohort to those analyzed during SAMBA. Their clinical acuity will have been assessed as less than those requiring inpatient care by a senior clinician.

2. The prediction in this cohort is thus predominantly not "admission" to hospital but actually a marker of "readmission". This should be the metric in this study.

3. The conclusion that the clinical acuity of those admitted to virtual wards is similar to those admitted on the acute medical take is thus deeply flawed.

.

Reviewer #1: No

---

## [Author Response · Author response to Decision Letter 1]

2 Dec 2025

Thank you so much for your incredibly detailed and helpful comments and suggestions. Please see point-by-point responses in the "Response to reviewer" document uploaded as an attached file. We hope our changes have addressed your concerns and improved the manuscript. Best wishes, Lauren Scott (on behalf of all authors)

---

## [Decision Letter · Decision Letter 1]

2 Feb 2026

Dear Dr. Scott,

Thank you for submitting your manuscript to PLOS ONE. After careful consideration, we feel that it has merit but does not fully meet PLOS ONE’s publication criteria as it currently stands. Therefore, we invite you to submit a revised version of the manuscript that addresses the points raised during the review process.

We look forward to receiving your revised manuscript.

Kind regards,

Tarek Samy Abdelaziz, MD,FRCP

Academic Editor

PLOS One

Journal Requirements:

**Additional Editor Comments:**

**Please carefully address the reviewers comments especially regarding missing data**

Reviewers' comments:

Reviewer's Responses to Questions

**Comments to the Author**

Reviewer #2: (No Response)

Reviewer #3: (No Response)

2. Is the manuscript technically sound, and do the data support the conclusions?

Reviewer #2: (No Response)

Reviewer #3: No

3. Has the statistical analysis been performed appropriately and rigorously?

Reviewer #2: Yes

Reviewer #3: No

4. Have the authors made all data underlying the findings in their manuscript fully available?

Reviewer #2: Yes

Reviewer #3: No

5. Is the manuscript presented in an intelligible fashion and written in standard English?

Reviewer #2: Yes

Reviewer #3: Yes

Reviewer #2: Rather than putting the median and IQR by demographics in the supplementary material the authors could have actually shown the distributions through histograms. That would have been more informative.

I have concerns about the large amount of missing data. The authors' response indicates that data at the beginning was unavailable because the full digitization had not been completed. If this is the case, I'm curious why then the study period was not adjusted to have only the time of full digitization? The authors are sort of mixing the study because they have patients that they describe for all variables and then some for only some variables. I do not quite feel the response to the reviewer's comment is satisfactory on that.

Reviewer #3: Thank you for letting me have the chance to see this work relating to an important aspect of care.

I have four major issues with the paper:

1.Missing data

The levels of missing data are, as described by Reviewer 1, almost insurmountable in terms of reliability of the findings. This is not least as there are clear imbalances in the degree of missingness, for example 23% of the youngest patients are excluded rising to 60% of the oldest patients, who are not only the largest age group but surely the group to most likely experience the outcomes of interest.

The authors give at least two explanations for this. They state that as a new service more data were missing at the earlier stages of the service but there is no clear evidence of any improvement over time in Figure 1 that suggest limiting the scope to a more recent period would improve the reliability of the analysis.

In the discussion its stated that the kit is not universally distributed and in fact the elderly (and likely more frail) are less likely to get it. The flip side of this then is surely that those at highest risk of the outcome were systematically excluded from this dataset and therefore this dataset is more representative of the lower risk patients than the entire virtual ward population as a whole.

At the least the paper needs to make the degree of missingness in all groups explicit in a table and beef up the discussion of the implications of this.

2.Only counting first admission

The rationale for, and implications of, using only the first virtual ward admission where there was more than one is not made clear and appears to create a number of important gaps in the data which aren’t mentioned, quantified, or addressed.

For example in this dataset, no patient can have both outcomes, they cannot both be re(admitted) to hospital and die as an outcome of their first virtual ward admission.

In reality of course a patient could have both, i.e. they get (re)admitted and die in hospital, but here only the readmission counts.

Another scenario which appears to be overlooked is where a virtual ward admission leads to a hospital (re)admission which leads to another virtual ward admission. The authors wording implies that this second virtual ward admission is not in the dataset and so any readmission or death at the end of that is not counted.

Whilst counting these deaths would be addressing a subtly different question (akin to the outcomes of a spell of care) it’s hard to see why that wouldn’t be a more important question than one which essentially ignores anything that happens after the first virtual ward admission.

As it is the analysis takes a person based approach, nobody can appear in the dataset more than once. And to facilitate that they use only the first virtual ward admission. It is not clear why the analysis of virtual ward admission outcomes could not be on an virtual ward admission basis, i.e. individuals could appear more than once.

Or with perhaps a stronger rationale than the first admission, to be based on the final virtual ward admission within a continuous spell, mirroring the approach used for hospital episodes within a spell. At least this would tell you the ultimate outcome of the spell of care, rather than just of part of it.

If the paper remains focused on the first virtual ward admission then it needs to quantify somewhere the outcomes for the omitted ward admissions, principally the final virtual ward admission. This would then enable the reader to understand if the first admission approach was suitable or not.

3.Composite outcome measure

The composite outcome measure of (re)admission or mortality (on the virtual ward) is problematic as clearly the implications to the two aspects are markedly different.

This exact combination of mortality or hospital admission is criticised in a BMJ paper on the deficiencies in composite outcome measures where the components are “not of similar importance” (doi: https://doi.org/10.1136/bmj.c3920).

This is further compounded by the decision to count only the first virtual ward admission as discussed above. Death, the most important outcome to the patient, is certain to be undercounted in the sense of final outcome at the end of the connected spell of care.

Whilst the stated number of deaths is small, this is certain to be an underestimate and it would be better to treat the two outcomes separately, having first identified the final outcome of the spell.

At the very least the paper should tabulate somewhere the number of outcomes by the demographic groups to give some sense of the risk and therefore help assess the impacts of the inherent biases in the dataset.

Ideally it would establish and focus on the final outcome of the “spell” of care.

4.Unadjusted analysis

I don’t really understand why just univariate analysis has been used.

It would seem that a multivariable approach would be more informative and may indicate important patterns not visible at the moment, for example perhaps the score is predictive in certain groups (perhaps those with less missing data) and not in others.

At the very least there needs to be a clear rationale stated as to why the current univariate analytical approach is the best approach to assessing predictiveness and give reassurance that there are no subgroup effects that have been overlooked.

.

Reviewer #2: No

Reviewer #3: No

---

## [Author Response · Author response to Decision Letter 2]

26 Feb 2026

Thank you for the opportunity to edit and resubmit our manuscript. Please see point-by-point responses to all reviewer comments in the "Response to Reviewer" document uploaded as an attached file.

---

## [Decision Letter · Decision Letter 2]

29 Mar 2026

Dear Dr. Scott,

Thank you for submitting your manuscript to PLOS ONE. After careful consideration, we feel that it has merit but does not fully meet PLOS ONE’s publication criteria as it currently stands. Therefore, we invite you to submit a revised version of the manuscript that addresses the points raised during the review process.

As the corresponding author, your ORCID iD is verified in the submission system and will appear in the published article. PLOS supports the use of ORCID, and we encourage all coauthors to register for an ORCID iD and use it as well. Please encourage your coauthors to verify their ORCID iD within the submission system before final acceptance, as unverified ORCID iDs will not appear in the published article. *Only* the individual author can complete the verification step; PLOS staff the individual author can complete the verification step; PLOS staff the individual author can complete the verification step; PLOS staff the individual author can complete the verification step; PLOS staff *cannot* verify ORCID iDs on behalf of authors.verify ORCID iDs on behalf of authors.verify ORCID iDs on behalf of authors.verify ORCID iDs on behalf of authors.

We look forward to receiving your revised manuscript.

Kind regards,

Tarek Samy Abdelaziz, MD,FRCP

Academic Editor

PLOS One

Journal Requirements:

Reviewers' comments:

Reviewer's Responses to Questions

**Comments to the Author**

Reviewer #2: All comments have been addressed

Reviewer #3: (No Response)

2. Is the manuscript technically sound, and do the data support the conclusions?

Reviewer #2: Yes

Reviewer #3: Yes

3. Has the statistical analysis been performed appropriately and rigorously?

Reviewer #2: Yes

Reviewer #3: Yes

4. Have the authors made all data underlying the findings in their manuscript fully available?

Reviewer #2: Yes

Reviewer #3: No

5. Is the manuscript presented in an intelligible fashion and written in standard English?

Reviewer #2: Yes

Reviewer #3: Yes

Reviewer #2: (No Response)

Reviewer #3: Thank you for your consideration of my comments and for the resulting changes you have made which make it generally much easier to read and interpret.

My only final comment is that you can see now in the new supplemental table 3 a gradient for hospital readmission risk by First NEWS2 score (NEWS2 0 = 5.2%, then 6.4%, 7.7%, 8.3% up to NEWS2 7+ = 12.7%), which surprisingly you don't mention in the actual paper and seems at face value to to run contrary to your claim that "...prediction of readmission to hospital using NEWS2 in this setting was poor".

Plugging those numbers into a quick Chi square confirms your finding that its not statistically significant but it does to me seem to warrant a mention. If your study was twice the size then the Chi square would be highly significant (and clinically meaningfully so as the risk in the 7+ group is 2.5 times that of the risk the 0 group).

It does seem to me that your paper would benefit from being a bit clearer about this as a weakness (there is evidence of a trend but the sample size is relatively small) rather than just saying its prediction ability was poor.

.

Reviewer #2: No

Reviewer #3: No

---

## [Author Response · Author response to Decision Letter 3]

30 Mar 2026

Thank you for the opportunity to edit and resubmit our manuscript. Please see our response to the final outstanding reviewer comment in the "Response to Reviewer" document uploaded as an attached file.

---

## [Editor Report · Decision Letter 3]

7 Apr 2026

Clinical acuity and National Early Warning Scores (NEWS2) of remotely monitored patients on virtual wards: a retrospective cohort study

PONE-D-25-43041R3

Dear Dr. Scott,

We’re pleased to inform you that your manuscript has been judged scientifically suitable for publication and will be formally accepted for publication once it meets all outstanding technical requirements.

Kind regards,

Tarek Samy Abdelaziz, MD,FRCP

Academic Editor

PLOS One
---

## [Editor Report · Acceptance letter]

PONE-D-25-43041R3

PLOS One

Dear Dr. Scott,

I'm pleased to inform you that your manuscript has been deemed suitable for publication in PLOS One. Congratulations! Your manuscript is now being handed over to our production team.

Kind regards,

on behalf of

Professor Tarek Samy Abdelaziz

Academic Editor

PLOS One